# A   Memory usage compared to 16-bit precision

Table 3 compares the memory footprint of 16-bit inference and LLM.int8() for different open source models. We can see, that LLM.int8() allows to run the largest open source models OPT-175B and BLOOM-176B on a single node equipped with consumer-grade GPUs.

Table 3: Different hardware setups and which methods can be run in 16-bit vs. 8-bit precision. We can see that our 8-bit method makes many models accessible that were not accessible before, in particular, OPT-175B/BLOOM.

| | | | Largest Model that can be run | |
|---|---|---|---|---|
| Class | Hardware | GPU Memory | 8-bit | 16-bit |
| Enterprise | 8x A100 | 80 GB | **OPT-175B / BLOOM** | **OPT-175B / BLOOM** |
| Enterprise | 8x A100 | 40 GB | **OPT-175B / BLOOM** | OPT-66B |
| Academic server | 8x RTX 3090 | 24 GB | **OPT-175B / BLOOM** | OPT-66B |
| Academic desktop | 4x RTX 3090 | 24 GB | **OPT-66B** | OPT-30B |
| Paid Cloud | Colab Pro | 15 GB | **OPT-13B** | GPT-J-6B |
| Free Cloud | Colab | 12 GB | **T0/T5-11B** | GPT-2 1.3B |

# B   Additional Related Work

**Quantization of Transformers with fewer than 1B Parameters**   Quantization of transformers has been focused on sub-billion parameter masked language model (MLMs), including BERT (Devlin et al., 2018) and RoBERTa (Liu et al., 2019). Versions of 8-bit BERT/RoBERTa include Q8BERT (Zafrir et al., 2019), QBERT (Shen et al., 2020), product quantization with quantization noise (Fan et al., 2020), TernaryBERT (Zhang et al., 2020), and BinaryBERT (Bai et al., 2021). Work by Zhao et al. (2021) performs both quantization and pruning. All these models require either quantization-aware finetuning or post-training quantization to make the model usable in low-precision. In contrast with our methods, the model can be used directly without performance degradation.

If one views matrix multiplication as 1x1 convolution, vector-wise quantization is equivalent to channel-wise quantization for convolution combined with row quantization (Khudia et al., 2021). For matrix multiplication, this was used by Wu et al. (2020) for BERT-sized transformers (350M parameters), while we are the first to study vector-wise quantization for autoregressive and large-scale models. The only other work that we are aware of that quantizes transformers other than BERT is Chen et al. (2020), which uses post-training quantization with zeropoint quantization in the forward pass and zeropoint-row-wise quantization in the backward pass. However, this work is still for sub-billion parameter transformers. We compare with both zeropoint and row-wise quantization in our evaluations and do not require post-training quantization.

**Low-bitwidth and Convolutional Network Quantization**   Work that uses less than 8-bits for data types is usually for convolutional networks (CNNs) to reduce their memory footprint and increase inference speed for mobile devices while minimizing model degradation. Methods for different bit-widths have been studied: 1-bit methods (Courbariaux and Bengio, 2016; Rastegari et al., 2016; Courbariaux et al., 2015), 2 to 3-bit (Zhu et al., 2017; Choi et al., 2019), 4-bits (Li et al., 2019), more bits (Courbariaux et al., 2014), or a variable amount of bits (Gong et al., 2019). For additional related work, please see the survey of Qin et al. (2020). While we believe that lower than 8-bit width with some performance degradation is possible for billion-scale transformers, we focus on 8-bit transformers that *do not* degrade performance and that can benefit from commonly used GPUs that accelerates inference through Int8 tensor cores.

Another line of work that focuses on convolutional network quantization is to learn adjustments to the quantization procedure to improve quantization errors. For example, using Hessian information (Dong et al., 2019), step-size quantization (Esser et al., 2019), soft quantization (Gong et al., 2019), mixed-precision via linear programming optimization (Yao et al., 2021), and other learned quantization methods (Zhang et al., 2018; Gholami et al., 2021).

Table 4: Summary statistics of outliers with a magnitude of at least 6 that occur in at least 25% of all layers and at least 6% of all sequence dimensions. We can see that the lower the C4 validation perplexity, the more outliers are present. Outliers are usually one-sided, and their quartiles with maximum range show that the outlier magnitude is 3-20x larger than the largest magnitude of other feature dimensions, which usually have a range of [-3.5, 3.5]. With increasing scale, outliers become more and more common in all layers of the transformer, and they occur in almost all sequence dimensions. A phase transition occurs at 6.7B parameters when the same outlier occurs in all layers in the same feature dimension for about 75% of all sequence dimensions (SDim). Despite only making up about 0.1% of all features, the outliers are essential for large softmax probabilities. The mean top-1 softmax probability shrinks by about 20% if outliers are removed. Because the outliers have mostly asymmetric distributions across the sequence dimension $s$, these outlier dimensions disrupt symmetric absmax quantization and favor asymmetric zeropoint quantization. This explains the results in our validation perplexity analysis. These observations appear to be universal as they occur for models trained in different software frameworks (fairseq, OpenAI, Tensorflow-mesh), and they occur in different inference frameworks (fairseq, Hugging Face Transformers). These outliers also appear robust to slight variations of the transformer architecture (rotary embeddings, embedding norm, residual scaling, different initializations).

| Model | PPL↓ | Params | Outliers | | Frequency | | | Top-1 softmax p | |
| | | | Count | 1-sided | Layers | SDims | Quartiles | w/ Outlier | No Outlier |
|---|---|---|---|---|---|---|---|---|---|
| GPT2 | 33.5 | 117M | 1 | 1 | 25% | 6% | (-8, -7, -6) | 45% | 19% |
| GPT2 | 26.0 | 345M | 2 | 1 | 29% | 18% | (6, 7, 8) | 45% | 19% |
| FSEQ | 25.7 | 125M | 2 | 2 | 25% | 22% | (-40, -23, -11) | 32% | 24% |
| GPT2 | 22.6 | 762M | 2 | 0 | 31% | 16% | (-9, -6, 9) | 41% | 18% |
| GPT2 | 21.0 | 1.5B | 2 | 1 | 41% | 35% | (-11, -9, -7) | 41% | 25% |
| FSEQ | 15.9 | 1.3B | 4 | 3 | 64% | 47% | (-33, -21, -11) | 39% | 15% |
| FSEQ | 14.4 | 2.7B | 5 | 5 | 52% | 18% | (-25, -16, -9) | 45% | 13% |
| GPT-J | 13.8 | 6.0B | 6 | 6 | 62% | 28% | (-21, -17, -14) | 55% | 10% |
| FSEQ | 13.3 | 6.7B | 6 | 6 | 100% | 75% | (-44, -40, -35) | 35% | 13% |
| FSEQ | 12.5 | 13B | 7 | 6 | 100% | 73% | (-63, -58, -45) | 37% | 16% |

## C  Detailed Outlier Feature Data

Table 4 provides tabulated data from our outlier feature analysis. We provide the quartiles of the most common outlier in each transformer and the number of outliers that are one-sided, that is, which have asymmetric distributions which do not cross zero.

## D  Inference Speedups and Slowdowns

### D.1  Matrix Multiplication benchmarks

While our work focuses on memory efficiency to make models accessible, Int8 methods are also often used to accelerate inference. We find that the quantization and decomposition overhead is significant, and Int8 matrix multiplication itself only yields an advantage if the entire GPU is well saturated, which is only true for large matrix multiplication. This occurs only in LLMs with a model dimension of 4096 or larger.

Detailed benchmarks of raw matrix multiplication and quantization overheads are seen in Table 5. We see that raw Int8 matrix multiplication in cuBLASLt begins to be two times faster than cuBLAS at a model size of 5140 (hidden size 20560). If inputs need to be quantized and outputs dequantized – a strict requirement if not the entire transformer is done in Int8 – then the speedups compared to 16-bit is reduced to 1.6x at a model size of 5140. Models with model size 2560 or smaller are slowed down. Adding mixed precision decomposition slows inference further so that only the 13B and 175B models have speedups.

These numbers could be improved significantly with optimized CUDA kernels for the mixed precision decomposition. However, we also see that existing custom CUDA kernels are much faster than when we use default PyTorch and NVIDIA-provided kernels for quantization which slow down all matrix multiplications except for a 175B model.

Table 5: Inference speedups compared to 16-bit matrix multiplication for the first hidden layer in the feed-forward of differently sized GPT-3 transformers. The hidden dimension is 4x the model dimension. The 8-bit without overhead speedups assumes that no quantization or dequantization is performed. Numbers small than 1.0x represent slowdowns. Int8 matrix multiplication speeds up inference only for models with large model and hidden dimensions.

| GPT-3 Size | Small | Medium | Large | XL | 2.7B | 6.7B | 13B | 175B |
|---|---|---|---|---|---|---|---|---|
| Model dimension | 768 | 1024 | 1536 | 2048 | 2560 | 4096 | 5140 | 12288 |
| FP16-bit baseline | 1.00x | 1.00x | 1.00x | 1.00x | 1.00x | 1.00x | 1.00x | 1.00x |
| Int8 without overhead | 0.99x | 1.08x | 1.43x | 1.61x | 1.63x | 1.67x | 2.13x | 2.29x |
| Absmax PyTorch+NVIDIA | 0.25x | 0.24x | 0.36x | 0.45x | 0.53x | 0.70x | 0.96x | 1.50x |
| Vector-wise PyTorch+NVIDIA | 0.21x | 0.22x | 0.33x | 0.41x | 0.50x | 0.65x | 0.91x | 1.50x |
| Vector-wise | **0.43x** | **0.49x** | **0.74x** | **0.91x** | **0.94x** | **1.18x** | **1.59x** | **2.00x** |
| LLM.int8() (vector-wise+decomp) | 0.14x | 0.20x | 0.36x | 0.51x | 0.64x | 0.86x | 1.22x | 1.81x |

## D.2  End-to-end benchmarks

Besides matrix multiplication benchmarks, we also test the end-to-end inference speed of BLOOM-176B in Hugging Face. Hugging Face uses an optimized implementation with cached attention values. Since this type of inference is distributed and, as such, communication dependent, we expect the overall speedup and slowdown due to Int8 inference to be smaller since a large part of the overall inference runtime is the fixed communication overhead.

We benchmark vs. 16-bit and try settings that use a larger batch size or fewer GPUs in the case of Int8 inference, since we can fit the larger model on fewer devices. We can see results for our benchmark in Table 6. Overall Int8 inference is slightly slower but close to the millisecond latency per token compared to 16-bit inference.

Table 6: Ablation study on the number of GPUs used to run several types of inferences of BLOOM-176B model. We compare the number of GPUs used by our quantized BLOOM-176B model together with the native BLOOM-176B model. We also report the *per-token* generation speed in milliseconds for different batch sizes. We use our method integrated into transformers(Wolf et al., 2019) powered by accelerate library from HuggingFace to deal with multi-GPU inference. Our method reaches a similar performance to the native model by fitting into fewer GPUs than the native model.

| Batch Size | Hardware | 1 | 8 | 32 |
|---|---|---|---|---|
| bfloat16 baseline | 8xA100 80GB | **239** | **32** | 9.94 |
| LLM.int8() | 8xA100 80GB | 253 | 34 | 10.44 |
| LLM.int8() | 4xA100 80GB | 246 | 33 | 9.40 |
| LLM.int8() | 3xA100 80GB | 247 | 33 | **9.11** |

## E  Training Results

We test Int8 training on a variety of training settings and compare to 32-bit baselines. We test separate settings for running the transformer with 8-bit feed-forward networks with and without 8-bit linear projections in the attention layer, as well at the attention iteself in 8-bit and compare against 32-bit performance. We test two tasks (1) language modeling on part of the RoBERTa corpus including Books (Zhu et al., 2015), CC-News (Nagel, 2016), OpenWebText (Gokaslan and Cohen, 2019), and CC-Stories (Trinh and Le, 2018); and (2) neural machine translation (NMT) (Ott et al., 2018) on WMT14+WMT16 (Macháček and Bojar, 2014; Sennrich et al., 2016).

The results are shown in Table 7 and Table 8. We can see that for training, using the attention linear projections with Int8 data types and vector-wise quantization leads to degradation for NMT and for 1.1B language model but not for 209M language modeling. The results improve slightly if mixed-precision decomposition is used but is not sufficient to recover full performance in most cases. These suggests that training with 8-bit FFN layers is straightforward while other layers require

additional techniques or different data types than Int8 to do 8-bit training at scale without performance degradation.

Table 7: Initial results on small and large-scale language modeling. Doing attention in 8-bit severely degrades performance and performance cannot fully recovered with mixed-precision decomposition. While small-scale language models is close to baseline performance for both 8-bit FFN and 8-bit linear projects in the attention layers performance degrades at the large scale.

| Params | Is 8-bit | | | Decomp | PPL |
| | FFN | Linear | Attention | | |
|---|---|---|---|---|---|
| 209M | | | | 0% | 16.74 |
| 209M | ✓ | | | 0% | 16.77 |
| 209M | ✓ | ✓. | | 0% | 16.83 |
| 209M | ✓ | ✓ | | 2% | 16.78 |
| 209M | ✓ | ✓ | | 5% | 16.77 |
| 209M | ✓ | ✓ | | 10% | 16.80 |
| 209M | ✓ | ✓ | ✓ | 2% | 24.33 |
| 209M | ✓ | ✓ | ✓ | 5% | 20.00 |
| 209M | ✓ | ✓ | ✓ | 10% | 19.00 |
| 1.1B | | | | 0% | 9.99 |
| 1.1B | ✓ | | | 0% | 9.93 |
| 1.1B | ✓ | ✓ | | 0% | 10.52 |
| 1.1B | ✓ | ✓ | | 1% | 10.41 |

# F   Fine-tuning Results

We also test 8-bit finetuning on RoBERTa-large finetuned on GLUE. We run two different setups: (1) we compare with other Int8 methods, and (2) we compare degradation of finetuning with 8-bit FFN layers as well as 8-bit attention projection layers comparel to 32-bit. We finetune with 5 random seeds and report median performance.

Table 9 compares with different previous 8-bit methods for finetuning and shows that vector-wise quantization improves on other methods. Table 10 shows the performance of FFN and/or linear attention projections in 8-bit as well as improvements if mixed-precision decomposition is used. We find that 8-bit FFN layers lead to no degradation while 8-bit attention linear projections lead to degradation if not combined with mixed-precision decomposition where at least the top 2% magnitude dimensions are computed in 16-bit instead of 8-bit. These results highlight the critical role of mixed-precision decomposition for finetuning if one wants to not degrade performance.

Table 8: Neural machine translation results for 8-bit FFN and linear attention layers for WMT14+16. Decomp indicates the percentage that is computed in 16-bit instead of 8-bit. The BLEU score is the median of three random seeds.

| Is 8-bit | | Decomp | BLEU |
| FFN | Linear | | |
|---|---|---|---|
| | | 0% | 28.9 |
| ✓ | | 0% | 28.8 |
| ✓ | ✓ | 0% | unstable |
| ✓ | ✓ | 2% | 28.0 |
| ✓ | ✓ | 5% | 27.6 |
| ✓ | ✓ | 10% | 27.5 |

Table 9: GLUE finetuning results for quantization methods for the feedforward layer in 8-bit while the rest is in 16-bit. No mixed-precision decomposition is used. We can see that vector-wise quantization improve upon the baselines.

| Method | MNLI | QNLI | QQP | RTE | SST-2 | MRPC | CoLA | STS-B | Mean |
|---|---|---|---|---|---|---|---|---|---|
| 32-bit Baseline | 90.4 | 94.9 | 92.2 | 84.5 | 96.4 | 90.1 | 67.4 | 93.0 | 88.61 |
| 32-bit Replication | 90.3 | 94.8 | 92.3 | 85.4 | 96.6 | 90.4 | 68.8 | 92.0 | 88.83 |
| Q-BERT (Shen et al., 2020) | 87.8 | 93.0 | 90.6 | 84.7 | 94.8 | 88.2 | 65.1 | 91.1 | 86.91 |
| Q8BERT (Zafrir et al., 2019) | 85.6 | 93.0 | 90.1 | 84.8 | 94.7 | 89.7 | 65.0 | 91.1 | 86.75 |
| PSQ (Chen et al., 2020) | 89.9 | 94.5 | 92.0 | **86.8** | 96.2 | 90.4 | 67.5 | 91.9 | 88.65 |
| Vector-wise | **90.2** | **94.7** | **92.3** | 85.4 | **96.4** | **91.0** | **68.6** | **91.9** | **88.81** |

Table 10: Breakdown for 8-bit feedforward network (FFN) and linear attention layers for GLUE. Scores are median of 5 random seeds. Decomp indicates the percentage that is decomposed into 16-bit matrix multplication. Compared to inference, fine-tuning appears to need a higher decomp percentage if the linear attention layers are also converted to 8-bit.

| Is 8-bit | | | | | | | | | | | |
|---|---|---|---|---|---|---|---|---|---|---|---|
| FFN | Linear | Decomp | MNLI | QNLI | QQP | RTE | SST-2 | MRPC | CoLA | STS-B | MEAN |
| | | 0% | 90.4 | 94.9 | 92.2 | 84.5 | 96.4 | 90.1 | 67.4 | 93.0 | 88.6 |
| ✓ | | 0% | 90.2 | 94.7 | 92.3 | 85.4 | 96.4 | 91.0 | 68.6 | 91.9 | 88.8 |
| ✓ | ✓ | 0% | 90.2 | 94.4 | 92.2 | 84.1 | 96.2 | 89.7 | 63.6 | 91.6 | 87.7 |
| ✓ | ✓ | 1% | 90.0 | 94.6 | 92.2 | 83.0 | 96.2 | 89.7 | 65.8 | 91.8 | 87.9 |
| ✓ | ✓ | 2% | 90.0 | 94.5 | 92.2 | 85.9 | 96.7 | 90.4 | 68.0 | 91.9 | 88.7 |
| ✓ | ✓ | 3% | 90.0 | 94.6 | 92.2 | 86.3 | 96.4 | 90.2 | 68.3 | 91.8 | 88.7 |