# OpenReview forum: "GPT3.int8(): 8-bit Matrix Multiplication for Transformers at Scale"
_NeurIPS.cc/2022/Conference — NeurIPS 2022 Accept_

### Official Review · Reviewer_2A6h · 2022-07-01

**Rating:** 6
**Confidence:** 4
**Soundness:** 3 good
**Presentation:** 3 good
**Contribution:** 3 good

**Summary:**

The paper proposes a method to quantize GPT style models using INT8 quantization. The paper finds that existing quantization methods perform poorly on quantizing GPT models. The authors identify that this was because of the outliers in feature maps, leading to large quantization errors. To address this issue, the authors propose to use finer-grained vector-wise quantization. The evaluation shows that the proposed method can quantize GPT style models without performance degradation to up to 13B parameters.

**Questions:**

Please cite and describe the relationship between this work and prior works on quantization, like those mentioned in the weaknesses section.

Please provide inference latency measurements or describe the hardware implication of the proposed method.

**Ethics Review Area:**

["I don’t know"]

**Limitations:**

Yes

**Strengths And Weaknesses:**

Strengths:

- The paper studies a timely problem and proposes practically useful techniques for quantizing large-scale GPT models.

- The paper evaluates the effectiveness of their approach on some very recent large-scale models such as billion-scale GPT-NeoX, OPT-30B, and demonstrates that they can be deployed on a single GPU with Int8 quantization.

Weaknesses:

- The paper misses quite some related work on quantization for transformer models, making it difficult to judge the novelty of the proposed method. For example, the outliers in feature maps have already been previously identified such as in "Understanding and Overcoming the Challenges of Efficient Transformer Quantization" by Bondarenko et al., https://arxiv.org/pdf/2109.12948.pdf. Therefore, the findings of emergent outliers on GPT models are not that new. Furthermore, the paper claims that its method does not require quantize-aware training, but again, the post-training quantization method has also been proposed by https://arxiv.org/pdf/2109.12948.pdf. Again, no references or discussions on this prior work. Finally, mixed-precision quantization and finer-grained quantization such as group-wise quantization have been studied in prior work such as https://www.stat.berkeley.edu/~mmahoney/pubs/NeurIPS-2020-hawq-v2.pdf, http://proceedings.mlr.press/v139/yao21a/yao21a.pdf, https://arxiv.org/pdf/1909.05840.pdf, https://www.ijcai.org/proceedings/2021/0472.pdf. Many of these works are either not cited or not properly discussed in the related work. Also, given that many of these techniques have been proposed and studied by prior works, the main contribution of the paper is limited to applying to billion-scale GPT models.

- The inference latency impact of the proposed method is not very clear. One of the main motivations for having INT8 quantization is that INT8 arithmetic computation is supported on commodity hardware such as Nvidia GPUs. Therefore, it appears to be important to either evaluate the inference latency of the proposed method or at least provide an explanation of the implication of the proposed method to the hardware in comparison to alternative quantization methods, e.g., those used together with QAT.


----------------
Post-rebuttal comments
The authors addressed my concerns. I increased my score from 5 to 6 to reflect my perspective after the rebuttal.

---

> ### Author Response · Authors · 2022-07-30
> **Addressing missing references and concerns; OPT-175/BLOOM inference results**
>
> Thank you for your careful review! You highlighted that evaluating large models and making them accessible is a strength of our work. Our updated version shows that our method also scales to OPT-175B and BLOOM-176B with no performance degradation, 1.95x memory reduction while maintaining inference speed.
>
> **The paper misses quite some related work [...] For example, the outliers in feature maps have already been previously identified such as in "Understanding and Overcoming the Challenges of Efficient Transformer Quantization"**
>
> Response: We will include this work in our related work discussion. While this work identifies outliers in residual connections, we find that outliers features become more and more systematic with scale, and more effective techniques can be developed once this relationship is understood. The emergent outliers we observe at scales beyond 6B parameters are not present in BERT models. While previous work shows performance degradation despite the handling of outliers, our method does not degrade performance even at the 175B parameter scale — a result that has not been achieved before in the literature.
>
> **Furthermore, [...] the post-training quantization method has also been proposed by https://arxiv.org/pdf/2109.12948.pdf. [...] Finally, mixed-precision quantization and finer-grained quantization such as group-wise quantization have been studied in prior work such as https://www.stat.berkeley.edu/~mmahoney/pubs/NeurIPS-2020-hawq-v2.pdf, http://proceedings.mlr.press/v139/yao21a/yao21a.pdf, https://arxiv.org/pdf/1909.05840.pdf, https://www.ijcai.org/proceedings/2021/0472.pdf.**
>
> Response: We will include this work in our discussion. We found that the first work that uses group-wise quantization for transformers is this one: “Integer quantization for deep learning inference: Principles and empirical evaluation”, which was also missing previously. We now reference all these works and relate them to our work which puts more emphasis on scaling to >=13B parameters without performance degradation. Our insights into systematic outliers and how to exploit them at the multi-billion scale is critical to this result.
>
> **[...] Also, given that many of these techniques have been proposed and studied by prior works, the main contribution of the paper is limited to applying to billion-scale GPT models.**
>
> Response: Our key method is mixed precision matrix decomposition for 8-bit matrix multiplication, which is novel and is key to our success as a scale. We believe our main contribution is best defined as the analysis of the emergence of systematic feature outliers and simple new methods for handling them to allow for zero-degradation Int8 quantization at scale. This contribution is novel, and no other work can claim 8-bit quantization without any performance degradation for multi-billion parameter models.
>
> **The inference latency impact of the proposed method is not very clear. [...]  it appears to be important to either evaluate the inference latency of the proposed method [...]**
>
> Response: This is similar to a request for latency measurements from reviewer JbRh. We copy our response to their similar concern verbatim:
> While the main concern of our work was to reduce the memory footprint of models, for example, we see a 1.96x reduction of memory for BLOOM-176B inference, we recognize that our method becomes worthless if it is too slow. As such, we run full end-to-end benchmarks to demonstrate that our method is just as fast as 16-bit inference — meaning quantization overhead and acceleration of 8-bit matrix multiplication roughly cancel out.
> We measure these numbers as follows: We load the BLOOM-176B checkpoint and then either use 8-bit or 16-bit (bf16) weights. We sample token-by-token with a batch size of 1, 8, and 32 sequences. We use a highly optimized implementation as a baseline (hugging face transformers + accelerate). Our results indicate that our method is as fast or faster than existing 16-bit inference (OOM indicates out of memory):
> |                          |              | Avg token inference time by batch size |      |        |
> | ------------------------ | ------------ | -------------------------------------- | ---- | ------ |
> | Method                   | Hardware     | 1                                      | 8    | 32     |
> | bf16 baseline            | 8x A100 80GB | 239ms                                  | 32ms | 9.9ms  |
> | bf16 baseline            | 4x A100 80GB | OOM                                    | OOM  | OOM    |
> | bf16 baseline            | 3x A100 80GB | OOM                                    | OOM  | OOM    |
> | Int8 (vectorwise+decomp) | 8x A100 40GB | 253ms                                  | 34ms | 10.4ms |
> | Int8 (vectorwise+decomp) | 4x A100 80GB | 246ms                                  | 33ms | 9.4ms  |
> | Int8 (vectorwise+decomp) | 3x A100 80GB | 247ms                                  | 33ms | 9.1ms  |

---

> > ### Comment · Reviewer_2A6h · 2022-08-08
> > **Thank you for the response**
> >
> > The authors addressed my concerns. The newly added results on OPT-175B/BLOOM-176B are encouraging. I will keep my result positive. The latency measurement also shows promising results, although the authors have been vague on what " highly optimized implementation" and "+accelerate" mean.

---

> > > ### Author Response · Authors · 2022-08-08
> > > **Additional information on inference setup**
> > >
> > > Thank you for your response and keeping your result positive. We also believe the no degradation results on OPT-175B/BLOOM-176B are very encouraging.
> > >
> > > Regarding the meaning of "+accelerate", we meant we use the Hugging Face Accelerate library: https://huggingface.co/docs/accelerate/package_reference/accelerator.
> > >
> > > By highly optimized implementation we meant:
> > > - cached attention context
> > > - optimized CUDA kernels for inference
> > >
> > > However, our implementation does not compile to ONNX runtime, which would be an addition step of optimization but which is currently difficult to realize with our custom implementation. ONNX runtime integration would be very complex and we believe our benchmarks are as fast as they could be for a non-ONNX implementation.

---

### Official Review · Reviewer_JbRh · 2022-07-11

**Rating:** 5
**Confidence:** 4
**Soundness:** 3 good
**Presentation:** 2 fair
**Contribution:** 3 good

**Summary:**

This paper explores how we can apply the uniform quantization method to large pre-trained language models. They utilize vector-wise quantization method and decomposition techniques for over 1B models and show the validation ppl of the 13B language model can be not degraded by uniform quantization method. For the 13B model, the decomposition method for outliers can improve the model performance regardless of quantization schemes.

**Questions:**

None

**Limitations:**

To support their assertions, I think this paper can be refined in two aspects: 1) additional experimental results on performance of generation models, and 2) additional arguments in the acceleration environment and implementation.

**Strengths And Weaknesses:**

<<Strengths>>
- Since there have been a few studies on large transformer models (over 1B parameters), the results on this paper can be a good reference for many researchers. We can learn the existence of many outliers in big transformer models (Table 2) and the performance degradations in conventional uniform quantization methods (Table 1).
- I think scaling to the 13B model is a big strength of this paper, and the consistent trend to 13B is also a good point of this paper.

<<Weaknesses>
- Although we can learn many things from Table 1, I have a concern on the experiments in the aspect of the limited evaluation. Many researchers have given attention to the powerful generation capability of Transformer decoders. But, this paper only measured the perplexities, the probabilistic results of language models. I think it seems to be not sufficient to note that there is no performance degradation on generation tasks. How about zero/few-shot evaluation results? With various kinds of tasks, the contribution of this paper would be strengthened.
- I have also another concern on the acceleration of this method, the vector-wise quantization and decomposition method. There is a short explanation about acceleration in Appendix A and Table 3. But I think it is not sufficient to prove there is no acceleration overhead. There is no description on the environmental setup of inference measurement. I think this paper should include some arguments about this issue in detail.

---

> ### Author Response · Authors · 2022-07-30
> **BLOOM/OPT-175B results and addressing concerns**
>
> Thank you for the time and diligence that you put into your excellent review! We like that you highlight the scale of our evaluation up to 13B parameters — this has been very difficult to do, but we believe it makes our results robust to scale. Indeed, as you requested, we evaluated on BLOOM-176B/OPT-175B, and found that we can also quantize these models with no degradation in both perplexity and zero-shot performance (as requested). We believe these results are very convincing since these are the largest public models. We address your main concerns in the next paragraphs. Please consider increasing your score if our response addresses your concerns.
>
> **[...] I have a concern on the experiments in the aspect of the limited evaluation [...] no performance degradation on generation tasks. How about zero/few-shot evaluation results? With various kinds of tasks, the contribution of this paper would be strengthened.**
>
> Response: From our experience, perplexity results generalize well to generation and zero-shot performance. To ensure our assumption is true, we evaluate zero-shot performance for the largest existing models OPT-175B and BLOOM-176B. We use the HF language model evaluation harness and use some of the most common tasks: hellaswag, Piqa, Lambada, Winogrande. From the results, we can see that our method does not degrade BLOOM/OPT zero-shot performance, which is in line with our perplexity results. We will fully flush out these evaluations and add them to the next version of the paper.
> | Model      | Bits | Hellaswag | Piqa  | Winogrande | Lambada | Lambada PPL | Avg Accuracy |
> | ---------- | ---- | --------- | ----- | ---------- | ------- | ----------- | ------------ |
> | OPT-175B   | 16   | 0.785     | 0.811 | 0.725      | 0.747   | 3.015       | 0.767        |
> | OPT-175B   | 8    | 0.785     | 0.810 | 0.717      | 0.746   | 3.014       | 0.765        |
> | BLOOM-176B | 16   | 0.730     | 0.791 | 0.705      | 0.672   | 3.931       | 0.725        |
> | BLOOM-176B | 8    | 0.727     | 0.792 | 0.705      | 0.681   | 3.919       | 0.726        |
>
> **[...] concern on the acceleration of this method, [...] not sufficient to prove there is no acceleration overhead. There is no description on the environmental setup of inference measurement. I think this paper should include some arguments about this issue in detail.**
>
> Response: The main goal of our work was to reduce memory use, for example, we see a 1.96x reduction of memory for BLOOM-176B inference. However, we recognize that our method becomes worthless if it is too slow. As such, we run full end-to-end benchmarks to demonstrate that our method is just as fast as 16-bit inference — meaning quantization overhead and acceleration of 8-bit matrix multiplication roughly cancel out.
> We measure these numbers as follows: We load the BLOOM-176B checkpoint and then either use 8-bit or 16-bit (bf16) weights. We sample token-by-token with a batch size of 1, 8, and 32 sequences. We use a highly optimized implementation as a baseline (hugging face transformers + accelerate). Our results indicate that our method is as fast or faster than existing 16-bit inference (OOM indicates out of memory):
> |                          |              | Avg token inference time by batch size |      |        |
> | ------------------------ | ------------ | -------------------------------------- | ---- | ------ |
> | Method                   | Hardware     | 1                                      | 8    | 32     |
> | bf16 baseline            | 8x A100 80GB | 239ms                                  | 32ms | 9.9ms  |
> | bf16 baseline            | 4x A100 80GB | OOM                                    | OOM  | OOM    |
> | bf16 baseline            | 3x A100 80GB | OOM                                    | OOM  | OOM    |
> | Int8 (vectorwise+decomp) | 8x A100 40GB | 253ms                                  | 34ms | 10.4ms |
> | Int8 (vectorwise+decomp) | 4x A100 80GB | 246ms                                  | 33ms | 9.4ms  |
> | Int8 (vectorwise+decomp) | 3x A100 80GB | 247ms                                  | 33ms | 9.1ms  |
>
> Please consider increasing your score if our response addresses your concerns.

---

> > ### Comment · Reviewer_JbRh · 2022-08-08
> > **Thank you for the response.**
> >
> > Thank you for additional results. It is a great job because the models are too big and rare to see the results of the biggest models.
> > - The first table about model performance can address my concern that there might be some sort of problems with zero/few-shot inference. (However, it doesn't make an effect on my score because they will add this results on the next paper.)
> > - For the second table (inference time), I really don't understand why the authors attach this table to argue their method. I pointed out that there could be acceleration overhead and it could lead to the damaged novelty of this paper. Unfortunately, this table including end-to-end testing cannot prove anything. Moreover, I don't understand why the BLOOM-176B model, not the models in this paper, is executed for proving. I'm so confused in front of this table. One of my questions is why the execution times are not quite different. The execution time must include network overhead (what is the type of network?), parallelism overhead (tensor parallel? pipelining?), other overheads, or matrix multiplication latency which we really want to measure. The latency time should be broken down and profiled. Or I cannot infer anything from this table. In addition, why is there no result on 8x A100 80GB GPUs for fair comparisons?

---

> > > ### Author Response · Authors · 2022-08-08
> > > **Clarifications and matrix latency benchmark numbers**
> > >
> > > Thank you for your timely response and for highlighting that the experimental results that we show for very large models is rare and valuable.
> > >
> > > I think we were not quite clear that we would add these results to the current paper. By next version we meant the version which we would use if our paper would get accepted. We would include the exact results that we added here in the camera ready version.
> > >
> > > For the second point, we misunderstood you. The latency comparisons for matrix multiplication vs 16-bit were already in the appendix, and we thought you asked for the end-to-end latencies. We reproduced the numbers from the appendix again in this response for your convenience. The table shows speedups compared to a 16-bit cuBLAS baseline (the latest CUDA version). You can see that there is a slowdown for small models, but for large models, there is a speedup if we look at raw matrix multiplication. This setup is for a batch size of 1 and a sequence length of 2048.
> > >
> > > |                                   | GPT-3 Size |        |       |      |      |      |      |       |
> > > | --------------------------------- | ---------- | ------ | ----- | ---- | ---- | ---- | ---- | ----- |
> > > |                                   | Small      | Medium | Large | XL   | 2.7B | 6.7B | 13B  | 175B  |
> > > | Model dim                         | 768        | 1024   | 1536  | 2048 | 2560 | 4096 | 5140 | 12288 |
> > > | 8-bit matmul with no quantization | 0.99       | 1.08   | 1.43  | 1.61 | 1.63 | 1.67 | 2.13 | 2.29  |
> > > | 8-bit with Vector-wise            | 0.5        | 0.5    | 0.75  | 0.91 | 1    | 1.18 | 1.61 | 2.03  |
> > > | 8-bit with Vector-wise+decomp     | 0.14       | 0.2    | 0.36  | 0.51 | 0.64 | 0.86 | 1.22 | 1.81  |
> > >
> > > It is difficult to do a full breakdown of the overall overhead since it depends highly on GPU, PCIe configuration, and the exact implementation. From the end-to-end benchmarks, we can see that matrix multiplication is only a small part of the overhead of inference due to the decreased speedups. As you correctly observe, likely, most of the overhead is due to networking. However, networking is also very hardware dependent. The quantization overhead can be estimated from the data in the table above, and for large models is about 10% for vector-wise and 20% for vector-wise combined with mixed precision decomposition.
> > >
> > > We chose BLOOM because it was the first experiment we ran for the rebuttal, and we ran speedup experiments along with it. For OPT models, we saw a rough speedup of 20% for the OPT-66B model for a networking setup with a single PCIe switch across 8 GPUs and 8x PCIe lanes per GPU. Speedups for OPT are likely larger for a better networking setup (our setup is optimized for CPU but not GPU networking). However, we did not record the data for all OPT models — mainly because these benchmarks are highly hardware dependent. We believe the numbers above are a better comparison as they can be replicated in a networking independent environment (just a single GPU).
> > >
> > > As such, we view the numbers for raw matrix multiplication above as the best comparison where we find relative difference in the performance of 8-bit and 16-bit matrix multiplication is roughly independent of the GPU as long as a Turing or Ampere GPU is used.

---

### Official Review · Reviewer_Booa · 2022-07-11

**Rating:** 5
**Confidence:** 4
**Soundness:** 3 good
**Presentation:** 3 good
**Contribution:** 2 fair

**Summary:**

This paper proposes a memory-efficient matrix multiplication method, mainly for multi-layer perceptron (MLP) and attention projection.
The method consists mainly of two parts.
One is vector-wise quantization to allow quantization with vector-wise independent scaling.
The other is mixed-precision matrix decomposition to adopt the outliers that often occur in larger Transformer models.
This method can reduce the required memory for inference by half while retaining full precision performance.
This paper first loads a 16/32-bit standard checkpoint and then converts it to Int8 for inference.
This method requires no post-quantization training.
This paper also reveals that existing quantization methods perform poorly at scale due to emergent outlier feature dimensions.
Experimental results on LM performance indicate that the result of the proposed method achieved no degradation from the original-size models.
These results indicate more accessibility to larger models (The authors state that GPT-J and T5-11B can work on a single free cloud GPU, GPT-NeoX-20B on a single gaming-grade GPU, and OPT-30B on a single data-center-grade GPU.



**Questions:**

* The definition of O, that is, $O = \\{i|i ∈ \\mathbb{Z}, 0 ≤ i ≤ h\\}$, in Section 2.2 is somewhat strange. Please check it again, and if this is not wrong, then could you show how to compute this set by using some actual examples.

* $ X^{h}\_{f16}$ and $ X^{h}\_{i8} $  in Eq. 8 should be in bold face.



**Limitations:**

This paper has limitations and potential negative social impact sections.


**Strengths And Weaknesses:**

Strengths:
* The existence of outlier features is already discussed in the previous study, but this paper seems to be the first paper that clearly indicates the reason for poor performance on existing quantization methods due to emergent outlier feature dimensions.
* The proposed method seems novel and unique.


Weaknesses:
* There are several missing definitions for notations in equations, and seems some errors (See Questions).
* The Absmax quantization and Zeropoint quantization seem not novel enough as main contribution of this paper since a similar approach has already been used in the literature (e.g., see https://intellabs.github.io/distiller/algo_quantization.html for example.
* The remaining part of the proposed method is Mixed-precision Matrix Decomposition. I feel that this technique is straightforward.

---

> ### Author Response · Authors · 2022-07-30
> **Addressing questions**
>
> Thank you for your time and for highlighting the discovery of the feature outliers as significant. We have recently applied the approach to OPT-175B/BLOOM-176B and found we can achieve zero performance degradation while regular quantization drops down to chance performance. This provides even more evidence that the insight of outlier feature dimensions is a highly important contribution. We address your individual concerns in turn:
>
>
> **“There are several missing definitions for notations in equations, and seems some errors”:**
>
> Response: Thank you for carefully checking the math! We will fix all the issues in the new draft. Regarding O={i|i∈Z,0≤i≤h}, we wanted to have both conditions be met, i∈Z and 0≤i≤h. We mistakenly used a comma instead of the symbol for and, ∧.
>
> **The Absmax quantization and Zeropoint quantization seem not novel enough.**
>
> Response: We agree that these approaches have been previously proposed, and are not claiming them as a novel contribution. We view absmax and zeropoint quantization as existing methods from the literature and believe that the comparison with zeropoint quantization is a critical baseline. Zeropoint quantization is not often used in practice, but in theory, its superior to absmax quantization in quantization precision. This is why we include zeropoint quantization in our comparison and include it in our background section. We do not claim that we developed this method.
>
> **The remaining part of the proposed method is Mixed-precision Matrix Decomposition. I feel that this technique is straightforward.**
>
> Response: The insight into how systematic and in which way outlier features emerge allows us to develop a simple method that works well in practice (e.g. to avoid the need for new complicated custom sparse matrix multiplication to deal with outliers). It is also faster in runtime and relatively easy to implement (no sparse algorithm required). As such, this insight helped us develop a fast and stable method for very large-scale models. As such, one of our main contributions — describing the systematic and emergent nature of outlier features at scale — also led to a simpler method. We believe the simplicity of our method is an advantage. This becomes clear with our recent update that showed this simple method generalized to BLOOM-176B/OPT-175B with zero-degradation of performance.
>
> If you found our response convincing please consider raising your score.

---

### Official Review · Reviewer_neVa · 2022-07-13

**Rating:** 7
**Confidence:** 4
**Soundness:** 3 good
**Presentation:** 3 good
**Contribution:** 3 good

**Summary:**

The largest language models in practice become resource intensive for inference which interferes with accessibility for researchers or e.g. edge device use cases. This paper offers a new type of weight quantization to int8 data type which enables lower overheads in inference for accessibility with reduced GPU utilization. Although int8 quantization is not new for language models, prior work has been interested in smaller models (e.g. on order of BERT scale).

One of the findings of this paper was that when such practices were attempted on larger scaled models (they looked at up to 13B parameters), there was an emergence of increased density of extreme outliers found within the weight values which impacted performance of prior quantization schemes. Their solution was to perform what they refer to as vector quantization, which scales the int8 quantization derivation unique to each vector after decomposing between those with prevalent outliers (which are retained as FP16) and those without (which are converted to int8), and after performing matrix multiplication the decomposed vectors are recombined into a common FP16 matrix. Their benchmarks demonstrated that the practice was able to duplicate performance of FP32 at up to 13B parameters, outperforming prior quantization schemes for high parameterization scales.

**Questions:**

1. Did the researchers consider or attempt to identify whether the emergence of extreme outliers in layers correlated with emergence of sparsity of weights? It seems possible that prior work noting sparsity of deeper models (e.g. Gissen et al, ICLR 2020) may have neglected to inspect for presence of extreme outliers, or for this work vice versa. Such alignment of sparsity and outliers emergence may suggest that it is a shared phenomenon.

2. The scaling methods of the paper appeared of most interest to negating potential for representation overflow in the quantization. Did author’s consider whether scaling based on a maximum value might result in a disproportionate degree of representation underflow? After all if scaling is based on extreme outliers, that suggests that a resulting absolute maximum scaled distribution will likely be “squished” into the lower registers of the data type. (Such distribution width is what is addressed when dividing by a standard deviation in z-score normalization, and wouldn’t be addressed by paper’s noted zero point extension.)

3. I am wondering if scaling derivations might be redundant with similar scaling performed as a part of batch normalization, and if so have authors considered a variant with hard coded scaling assumptions assuming presence of batch norm?

4. Could the authors please clarify what is meant by “feature dimensions h” (247) and how it is derived in comparison to outlier feature dimensions h_i? (I didn’t understand your example of “dimension i=888 for h=1024” (236))

5. I would have been interested in additional detail for context of the noted parameterization thresholds. While I recognize that this is a quantization process for inference, I expect the presence of outlier values arises during training, and various phenomenon from overparameterization (like double descent) are found to arise by parameterization in context of scale of training tokens. (See e.g. Belkin et al 2019 "Reconciling modern machine-learning practice and the classical bias–variance trade-off"). I would be curious if future work may yet identify that the benefit of this form of quantization, as well as the emergence of “outlier feature dimensions”, were found to be impacted by variations of such conditions experienced during training like proximity to the interpolation threshold (which is a function of both parameterization and training sample counts).

6. Does the method only work on low rank tensors as per the demonstration or can it be adapted to diverse configurations?

**Limitations:**

Ideally benchmarking could be extended to larger parameterizations exceeding 13B, the authors noted the possibility of additional emergence of feature dimensions. Since this is a method for inference as opposed to training authors may find that such experiments may not be hugely out of reach from a resource utilization standpoint.

**Strengths And Weaknesses:**

Strengths

1. Prior language model quantizations required adaptions during training, this approach can be performed in inference independent of training regime.

2. Prior language model quantizations experienced performance degradations a higher tiers of parameterization. This paper demonstrated through benchmarking that inference could be quantized without material impact to performance, demonstrated at scales up to 13B parameters.

3. These methods could make large language models much more accessible to research and industry by way of lowering GPU overhead in inference.


Weaknesses

1. Authors note that the method wasn’t applied to attention layers. Paper should have provided better detail of why this method was insufficient, especially considering that the intended use case is for large language models.

2. The paper introduced some interesting emergent phenomenon at larger parameterization scale (of outlier weight values) but did not attempt to draw context towards other phenomenon found in overparameterization.

3. I found the reference to what author’s call “outlier feature dimensions” lacked clarity of definition.

---

> ### Author Response · Authors · 2022-07-30
> **Answering questions; additional experimental results at the 175B scale.**
>
> Thank you for your excellent review highlighting our strengths of being able to quantize any model, in particular large-scale models, to 8-bit and use it immediately without performance degradation.
>
> **Authors note that the method wasn’t applied to attention layers. [...]**
>
> Response: We do not focus on the attention layers since they do not use any parameters and our main focus is to reduce the memory footprint for transformers. We still quantize both key/value/query and output attention projection. Even though we do not perform quantized attention for BLOOM-176B, for example, we quantize 98% of all parameters.
>
> **The paper introduced some interesting emergent phenomenon at larger parameterization scale (of outlier weight values) but did not attempt to draw context towards other phenomenon found in overparameterization.**
>
> Response: The number of outlier features in a transformer monotonically increases as the validation perplexity decreases. This relationship does not hold if the parameter count alone is considered. As such, this means, that both overparameterization and the number of training tokens processed lead explain the outlier features well, but neither of the variables alone.
>
> **I found the reference to what author’s call “outlier feature dimensions” lacked clarity of definition.**
>
> Response: We offer the following definition of outlier feature dimensions.
> Given a transformer with n layers where each layer produces a hidden state H of size [s, h] where s is the sequence dimension and h the hidden/feature dimension, an outlier feature dimension is a particular dimension j, H_{i, j}, where at least 5% of sequence dimensions for each hidden states in [0, s] and 25% of hidden states in [0, n] contain a value with magnitude >=6. An outlier feature dimension has the property that as language modeling perplexity decreases, the proportion of hidden states affected in [0, n] reaches 100%. Once 100% of hidden states are affected, we call the feature dimension emergent.
>
> **Did the researchers consider or attempt to identify whether the emergence of extreme outliers in layers correlated with emergence of sparsity of weights? [...]**
>
> Response: We indeed noted relationships, but these patterns are not reliable enough to make a definitive statement about sparsity. We observed the following patterns:
> - Attention gets sparser the more outlier features are present
> - Attention gets extremely sparse once the “emergence” threshold around 6.7B parameters is reached
> - FFN layers are more difficult to sparsify after “emergence” at around 6.7B parameters. Removing as few as the smallest 5% weights leads to significant performance degradation.
>
> **[...] Did author’s consider whether scaling based on a maximum value might result in a disproportionate degree of representation underflow? [...]**
>
> Response: This is indeed the correct interpretation. We can see this is correct from our zeropoint vs absmax quantization analysis. However, with mixed precision decomposition, the advantage disappears, showing that it is no longer a problem.
>
> **I am wondering if scaling derivations might be redundant with similar scaling performed as a part of batch normalization, and if so have authors considered a variant with hard coded scaling assumptions assuming presence of batch norm?**
>
> Response: Since layer normalization normalizes the feature dimension, it can be seen as redundant to row-wise quantization. Vector-wise quantization can add additional “normalization” to the weight. We find that vector-wise quantization is still very helpful in many cases, indicating that a re-parameterization of layer norm alone would not yield as good results as we demonstrate.
>
> **Could the authors please clarify what is meant by “feature dimensions h” (247) and how it is derived in comparison to outlier feature dimensions h_i? (I didn’t understand your example of “dimension i=888 for h=1024” (236))**
>
> Response: Here i is an index and h is the feature dimension. So a transformer with a feature / model dimension of 1024 h_888 would be the 888th dimension of the feature map / hidden state.
> Programmatically putting this, if the hidden state has a batch, sequence, and hidden dimension,  state = torch.rand(b, s, h), then h_888 represents the operation state[:, :, 888].
>
> **I would have been interested in additional detail for context of the noted parameterization thresholds. [...] I expect the presence of outlier values arises [...] from overparameterization [...] scale of training tokens. [...]**
>
> Response: Please see the response about overparameterization above. In short, we see evidence that outliers are related to both parameter count and the number of training tokens.
> Secondly, we find that the average magnitude of outlier features slowly but steadily increases until the emergence threshold. Once the threshold is reached, the outlier features grow by a factor of 15x.
>
> Please consider raising your score if you found our response convincing.

---

> > ### Comment · Reviewer_neVa · 2022-08-06
> > **Thank you for feedback**
> >
> > Thank you for your detailed feedback. In hindsight I see that some of my questions were not directly aligned to the main focus of the paper and appreciate your humoring some of what may have been better suited for a poster session q and a. I do consider this work a strong contribution and hope current scores reflect that.

---

> ### Author Response · Authors · 2022-07-30
> **Additional results**
>
> **Ideally benchmarking could be extended to larger parameterizations exceeding 13B [...]**
>
> Response: We used our method on BLOOM-176B and OPT-175B and find our method generalizes to these models indicating no additional emergent properties that affect quantization. We achieve zero-degradation quantization at this scale.
>
> | Model      | Bits | Hellaswag | Piqa  | Winogrande | Lambada | Lambada PPL | Avg Accuracy |
> | ---------- | ---- | --------- | ----- | ---------- | ------- | ----------- | ------------ |
> | OPT-175B   | 16   | 0.785     | 0.811 | 0.725      | 0.747   | 3.015       | 0.767        |
> | OPT-175B   | 8    | 0.785     | 0.810 | 0.717      | 0.746   | 3.014       | 0.765        |
> | BLOOM-176B | 16   | 0.730     | 0.791 | 0.705      | 0.672   | 3.931       | 0.725        |
> | BLOOM-176B | 8    | 0.727     | 0.792 | 0.705      | 0.681   | 3.919       | 0.726        |

---

### Meta-Review · Area_Chair_CWMK · 2022-08-28

**Recommendation:** Accept
**Confidence:** Certain

**Metareview:**

This paper provides a solid contribution to compressed ML by showing how to apply 8-bit arithmetic to large-scale transformers. All the reviewers had a borderline or positive impression of the paper, with agreement among the reviewers that the results would be interesting to
the community especially inasmuch as they represent an empirical exploration of compression on much larger scale models than previous studies. The result about the phase transition is particularly interesting, and may merit future work. While a couple of the reviewers assigned borderline scores, the weaknesses pointed out in the reviews mostly coalesced around questions of novelty and comparison to the literature: while I agree that novelty is important, I think that identifying a simple way to combine and adapt some subset of existing methods to work at new scales _is_ itself a novel contribution. (The authors should, of course, make sure to cite the additional papers mentioned by the reviewers as proposed in their author response.) And the author response does a good job of contextualizing the reviewers' concerns: there is no unaddressed weakness that would amount to a strong justification for rejection. At the same time, all reviewers seem to agree on the strengths of the paper, and this sort of result will be interesting and useful to the community. As such, I recommend acceptance.

**Award:**

No

---

### Decision · Program_Chairs · 2022-09-14

Accept